# Investigation of Time-Lapse Changes with DAS Borehole Data at the Brady Geothermal Field Using Deconvolution Interferometry

Hilary Chang [1,*,†] and Nori Nakata [1,2,†]

1   Massachusetts Institute of Technology, 77 Massachusetts Avenue, 54-216, Cambridge, MA 02139, USA; nnakata@mit.edu
2   Lawrence Berkeley National Laboratory, 1 Cyclotron Road, Berkeley, CA 94720, USA
*   Correspondence: hilarych@mit.edu
†   These authors contributed equally to this work.

**Abstract:** Distributed acoustic sensing (DAS) has great potential for monitoring natural-resource reservoirs and borehole conditions. However, the large volume of data and complicated wavefield add challenges to processing and interpretation. In this study, we demonstrate that seismic interferometry based on deconvolution is a convenient tool for analyzing this complicated wavefield. We also show the limitation of this technique, in that it still requires good coupling to extract the signal of interest. We extract coherent waves from the observation of a borehole DAS system at the Brady geothermal field in Nevada. The extracted waves are cable or casing ringing that reverberate within a depth interval. These ringing phenomena are frequently observed in the vertical borehole DAS data. The deconvolution method allows us to examine the wavefield at different boundary conditions and separate the direct waves and the multiples. With these benefits, we can interpret the wavefields using a simple 1D string model and monitor its temporal changes. The velocity of this wave varies with depth, observation time, temperature, and pressure. We find the velocity is sensitive to disturbances in the borehole related to increasing operation intensity. The velocity decreases with rising temperature. The reverberation can be decomposed into distinct vibration modes in the spectrum. We find that the wave is dispersive and the fundamental mode propagates with a large velocity. This interferometry method can be useful for monitoring borehole conditions or reservoir property changes using densely-sampled DAS data.

**Keywords:** distributed acoustic sensing; borehole; time-lapse

## 1. Introduction

Fiber-based sensors have been applied in the oil and gas industry for borehole monitoring since early 1990s [1]. Since then, distributed temperature sensors (DTS) have been routinely deployed for monitoring well temperatures. Distributed acoustic sensors (DAS) have gained popularity in seismology more recently. DAS measure strain rate, and thus, record the seismic wavefield as a string of one-component geophones. Advances in fiber material and computer technologies allow us to obtain higher quality data and analyze this data with array processing techniques.

DAS has been used in boreholes environments for a variety of applications. These include flow monitoring [2–4], wellbore diagnostics [2,4], vertical seismic profiling VSP; [5–8], hydraulic fracture characterization [9,10], and microseismicity detection [11]. DAS is suitable for borehole monitoring for several reasons [12,13]: First, DAS fiber has higher endurance in high temperature, high pressure, and corrosive environments compared to geophones. Second, it provides dense 1D receiver arrays along the wellbore. Finally, the cost of DAS borehole deployment is relatively low, although the interrogator and data storage can be expensive. Once installed, the fiber can be left in the well for

long-term monitoring without changing locations. This resolves one of the main difficulties for conventional 4D (3D and time) surveys.

A big challenge of analyzing DAS wavefields is that they are often complicated, especially in the borehole environment. Transient borehole processes such as fluid flows and operation activities cause disturbances in the borehole. Optical noises from the DAS interrogator create noise that occurs simultaneously and at the same amplitude on all sensor channels [5]. For the DAS data we analyze in this study, cable and casing ringing populate a large portion of the data [14]. Ringing is a common phenomenon for DAS in a vertical borehole due to poor coupling between the DAS cable and the casing, or between the casing and the formation [15–19]. It appears as bouncing waves that reverberate within a depth interval. For VSP applications, the ringing is a noise that analysts want to eliminate [20]. Here, we treat these ringing waves as signals and analyze their time-lapse changes. This allows us to interpret the dominant energy sources in the system and understand if the cable and the casing are sensitive to certain processes.

The DAS data we analyze are from a vertical borehole at the Brady geothermal field in Nevada. The data were obtained during the PoroTomo project [21,22]. The PoroTomo project was a four-week experiment conducted during March 2016 in which the team performed vibroseis experiments under varying pumping operations and collected a variety of geophysical data including surface DAS (DASH), borehole DAS (DASV), nodal geophones, InSAR, GPS, pressure, and temperature (DTS) data. The DASV data were available from 18–26 March 2016. Previous studies have analyzed the DASV, DTS, and pressure data. Patterson et al. [23] and Patterson [24] analyzed the borehole DTS and pressure data at different stages of operations. Trainor-Guitton et al. [25] imaged features on two nearby steeply dipping faults using a portion of the DASV data. Miller et al. [14] investigated the DASV data to find the signatures of earthquakes, vibroseis sweeps, and responses to different borehole processes. In addition, they suggested that reverberations on the upper half of the DASV are due to ringing of the casing and the DAS cable. We follow their results and further investigate the time-lapse changes of these reverberations.

We use deconvolution seismic interferometry to extract coherent signals along the 1D receivers of the borehole DASV array. The coherent signals are governed by the same wave physics (i.e., wave equation) [26]. Thus, we can understand the property of the structure by examining this wave. This deconvolution method is useful because it modifies the boundary conditions [26–28]. Thus, we can convert the wavefield to a favored boundary condition for interpretation. For example, Snieder and Safak [29], Nakata et al. [28], and Nakata and Snieder [30] used this method to isolate the ground coupling effect and analyzed the vibration modes of a building. Sawazaki et al. [31], Yamada et al. [32], Nakata and Snieder [33], and Bonilla et al. [34] applied similar methods to obtain near-surface velocity changes in different time scales. In this study, we use this deconvolution method to help us examine the wavefield. This allows us to interpret the wavefield using a simple model. Furthermore, it separates the direct waves and the multiples and simplifies the wavefields. This makes time-lapse monitoring easier to implement.

In the following sections, we first introduce the Brady DASV data (Section 2) and the deconvolution interferometry method (Section 3.1). We focus on analyzing the ringing signals on the shallower part of the borehole as their energies are consistent. In Section 3.2, we show the deconvolved wavefields and how they can be explained by our proposed models. Detailed parameters for modeling are given in Appendix A. In Section 3.3, we analyze the velocity variations of the signal versus measured depth, observation time, temperature, and pressure. In Section 3.4, we apply a normal-mode analysis to the vibration modes of the waves. We also show the signals we extracted from waves propagating in the formation in Appendix B. For these signals, we could not obtain precise velocity changes due to poor coupling. However, we note that if the coupling were better, similar techniques could be applied to these signals for analyzing temporal changes in the formation.

## 2. Data

We focus on the DASV, the DTS temperature, and the pressure data from the PoroTomo project [21,22]. Figure 1 shows the location of the wells relative to the entire DASH array and vibroseis shots on the surface. The DASV and DTS fibers are co-located in Well 56-1 (the red star), which is about 380 m deep. The DASV fiber is single-mode and the DTS fiber is multi-mode. Both fibers are high-temperature acrylate coated and are tested to be resilient up to 150 °C. For resistance, the fibers are protected by stainless steel double tubing. The DASV system has 384 channels with channel spacing of approximately 1 m. The gauge length is 10 m. The sampling rate is 1000/s. The unit of the DAS raw data is radian/millisecond per gauge length. The total DASV data size is 981 GB stored in SEG-Y format. The DTS system has channel interval of 0.126 m and the sampling interval is 62 s. The pressure sensor (P sensor) is located at a nearby well 56A-1 (the green dot). The pressure sensor is at an elevation corresponding to channel 219 of the DASV system (i.e., measured depth = 219 m). The sampling interval of the pressure sensor is 60 s. The two wells are around 100 m away from each other. Their wellheads are both at about the same elevation above sea level (1230 m). Patterson [24] suggested that the two wells are hydraulically connected based on simultaneous responses between the DTS and the pressure sensor. Hence, we assume the pressure measurements can represent the co-located pressure changes with the DASV and DTS at the measured depth of 219 m.

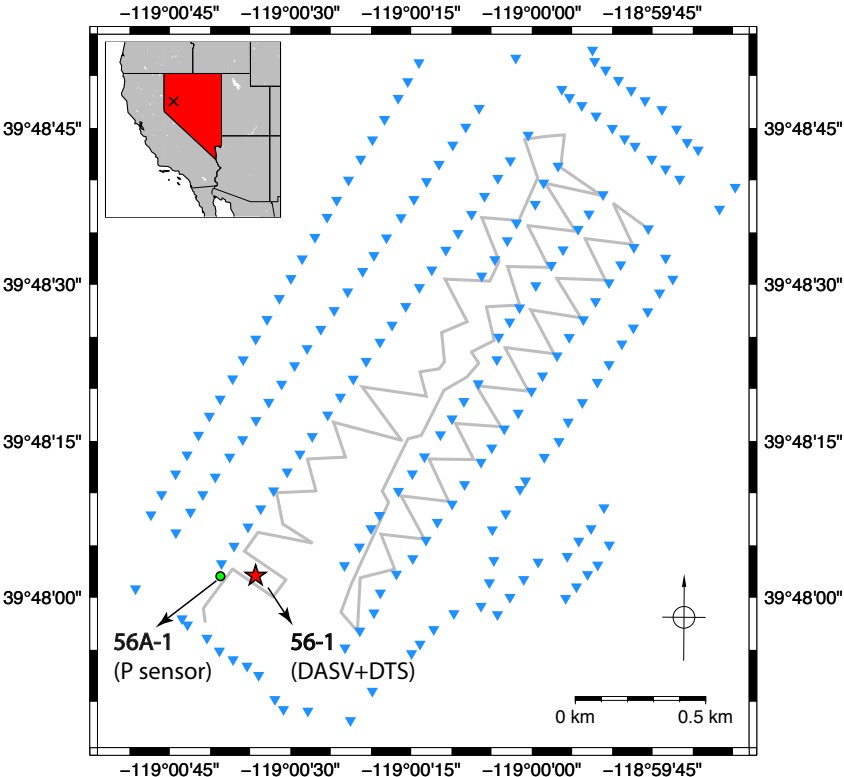

**Figure 1.** Locations of the target boreholes in the PoroTomo experiment. The survey was at the Brady geothermal field in Nevada, USA (black cross in the inset). The red star is the borehole with DASV and DTS (Well 56-1). The green dot is the borehole with the pressure (P) sensor (Well 56A-1). The blue triangles are locations of the vibroseis shots. The gray lines are the DASH cables on the surface. We use DASV, DTS, and pressure data in this study.

Figure 2 shows the evolution of the pressure and temperature, and an overview of the DASV DC values and root-mean-square (RMS) amplitudes. We focus on the eight days (18–26 March) where the DASV was actively recording. Initially on 18 March, the pressure dropped drastically due to the well resuming operation after a shutdown period (yellow to blue shade in Figure 2a). Then, the pressure increased slowly due to increasing injection,

until resuming normal operation on 24 March (blue to green shade). The sudden pressure rise at the end of 25 March is due to a plant shutdown [35]. The temperature increases with depth, with a heat deficit below 320 m due to geothermal explorations (Figure 2b; Miller et al. [14]). The lower temperature early on 18 March is due to cool water treatments before cable installation. Figure 2c,d show that the DASV data contains many disturbances under these changing pressure and temperature conditions. Patterson et al. [23] and Miller et al. [14] investigated these events. Here, we analyze the changes of extracted waves in the deconvolved wavefields.

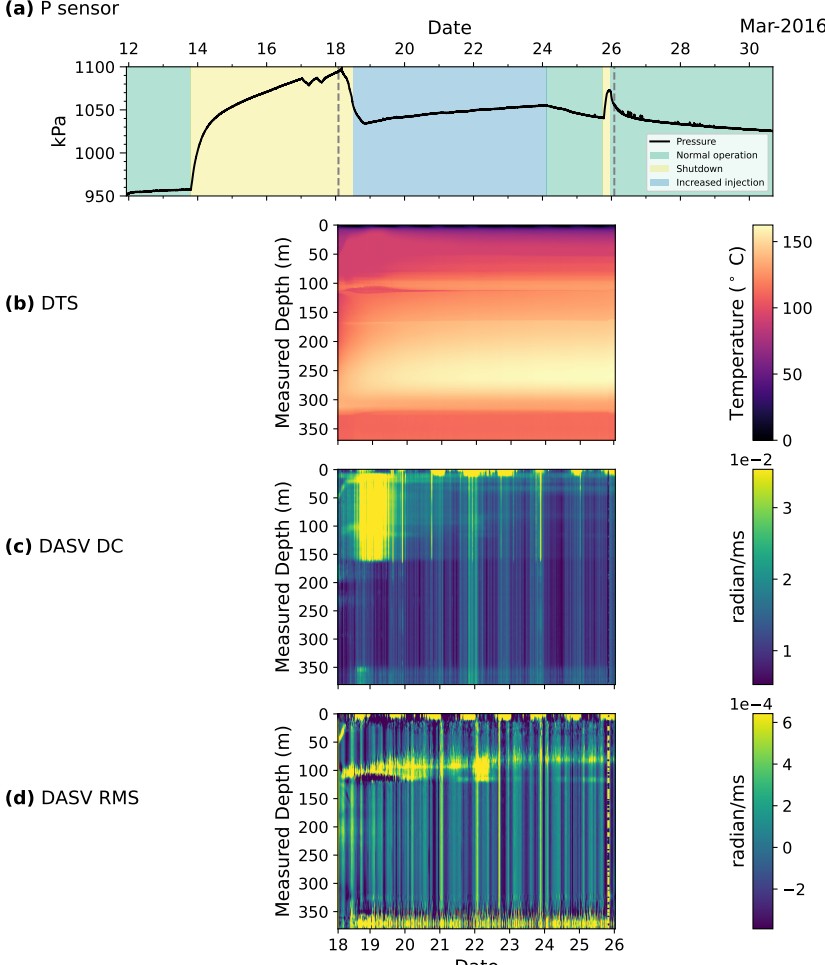

**Figure 2.** An overview of the pressure, temperature, and DASV data. (**a**) Pressure (P) and corresponding field operation stages. The analysis period of this study is 18–25 March (a total of eight days; marked by gray dashed lines). (**b**) Temperature profile from the DTS. (**c**) DASV DC amplitudes. (**d**) DASV root-mean-square (RMS) amplitudes. The DC and RMS amplitudes are calculated using a 30 min time window with 50% overlap.

## 3. Methods and Analysis

### 3.1. Review of Deconvolution Interferometry

We use deconvolution interferometry to extract coherent waves from the data. The receiver used for deconvolution is a "virtual source". The deconvolution operation modifies the boundary conditions of the wavefield depending on the virtual source [26–28]. These boundary conditions include coupling, attenuation, and damping at the boundaries that obscure the pure response of the system. By examining the wavefields that satisfy different boundary conditions, we can potentially separate these unwanted effects. For time-lapse monitoring, this allows us to track the pure response of the structure. Snieder and Safak [29] and Nakata et al. [28] used this deconvolution method to retrieve

the vibration modes of the building with receivers deployed along the building floors. Nakata and Snieder [33] monitored monthly and annual changes in shear-wave velocity using near-surface and borehole sensors. Sawazaki et al. [31], Yamada et al. [32], Nakata and Snieder [36], and Bonilla et al. [34] analyzed near-surface velocity changes during earthquake strong ground motions. Here, we use deconvolution interferometry to analyze the reverberations that are commonly observed for DAS in a vertical borehole. We also show its potential for time-lapse borehole condition and reservoir monitoring.

The deconvolved wavefield $D$ in the frequency domain is [28]:

$$D(z, z_a, \omega) = \frac{U_z(\omega)}{U_{z_a}(\omega)} \tag{1}$$

$$\approx \frac{U_z(\omega) U_{z_a}^*(\omega)}{|U_{z_a}|^2 + \varepsilon \langle |U_{z_a}|^2 \rangle} \quad, \tag{2}$$

where $z$ is the depth of each channel, $z_a$ is the depth of the virtual source channel, $\omega$ is the angular frequency, and $^*$ denotes the complex conjugate. The deconvolution operation in the frequency domain is the division of the data recorded at each depth ($U_z(\omega)$) by the data recorded by the receiver that is used as the virtual source ($U_{z_a}(\omega)$). The instability in Equation (1) comes from the division, and we stabilize it with a water level $\varepsilon = 0.5\%$ that scales with the average power spectrum $\langle |U_{z_a}|^2 \rangle$ in Equation (2). Given a virtual source channel, we calculate the deconvolved wavefield using Equation (2) for the entire 1D array and stack the resulting wavefields over a time span to improve the signal-to-noise ratio. Figure 3 summarizes the workflow for calculating the deconvolved wavefield.

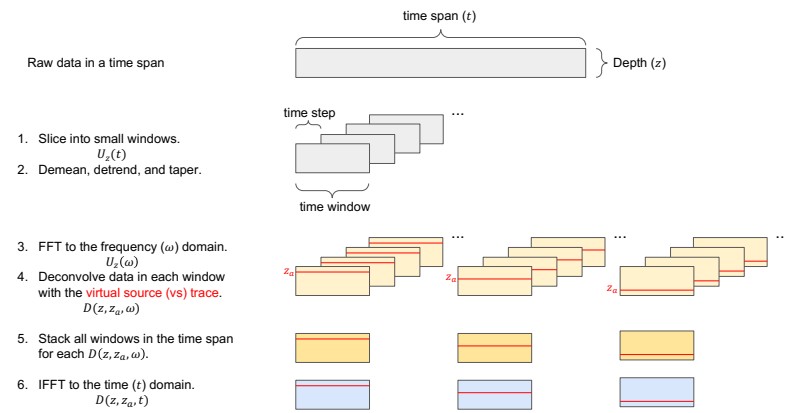

**Figure 3.** Flowchart for calculating the stacked deconvolved wavefields. For a given time span, we have five steps: (1) slice the data into small overlapping windows; (2) demean, detrend, and taper both ends in time; (3) use fast Fourier transform (FFT) to convert the data in each window into the frequency domain; (4) calculate the deconvolved wavefield using Equation (2) for each virtual source (vs.) trace at $z_a$; (5) stack the deconvolved wavefields of each virtual source; (6) use inverse FFT (IFFT) to convert the stacked deconvolved wavefields back to the time domain.

We use two sets of time windows to calculate the deconvolved wavefields. In Section 3.2, we use 30 min time windows and 50% time overlap (time step = 15 min), and then stack the deconvolved wavefields over 3 h (time span = 3 h) to enhance the signal-to-noise ratio. In Section 3.3, we use 1 min time windows and 50% time overlap (time step = 30 s), and then stack the deconvoluted wavefields over 1 h (time span = 1 h). Since the deconvolution is conducted in frequency domain Equation (2), we demean, detrend, and taper (10% on both sides) the raw data at each window before the Fourier transform. For simplicity, we omit the "stacked" term and call the final retrieved wavefield the "deconvolved wavefield" for the rest of this paper.

### 3.2. Deconvolved Wavefields

Figure 4 shows the deconvolved wavefields in the upper half of the borehole between 0–200 m (the top panels in subfigure a–f). We obtain strong reverberating signals that bounce between 10 and 165 m. They are only present when the virtual source is within the same depth interval. When we put the virtual source below 200 m, the reverberations almost disappear. This suggests that these waves are restricted in this depth interval. Figure 4a–c are wavefields in the same time window, but are deconvolved with different virtual sources marked by the red dash lines. They show distinct differences. The waves in the deconvolved wavefield are coherent energies, assuming that they are excited at the virtual source.

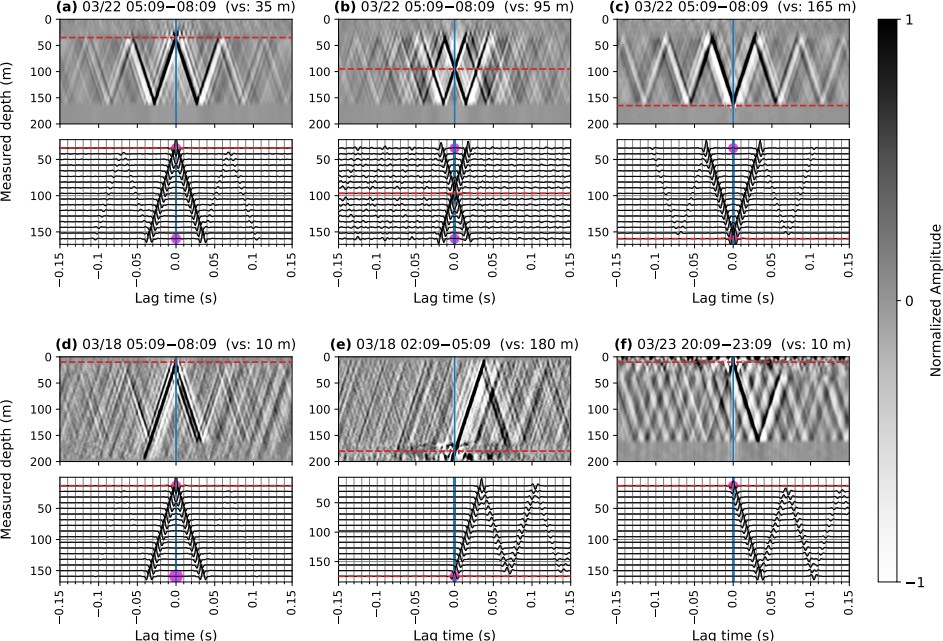

**Figure 4.** Comparisons between the DASV deconvolved wavefields and the simulated wavefields using Model 1. The deconvolved wavefields are calculated using 30 min time windows and 50% overlap, and are stacked over 3 h. The blue lines are the zero-lag times. The red dashed lines are the virtual source channels, also indicated by *vs* in the subtitles. In the simulated wavefields, the magenta balls mark where we put the actual sources. The size of the balls is proportional to the source amplitude. In (**a–c**), we examine the same wavefields using different virtual sources. The wavefields exhibit symmetry. In (**d–f**), we show cases of occasional variations. See text for details.

To explain the observed deconvolved wavefields, we use a simple string model and derive its mathematical notation. Figure 5 shows the sketch of the model (Model 1). This string model has two reflectors ($R_1$ and $R_2$) on the top and the bottom as boundaries, and two sources ($S_1$ and $S_2$) at those boundaries. This model can also represent the case in which there are sources that are further away from the end points outside of this receiver line [33]. Hence, one should consider $S_1$ and $S_2$ as the incoming waves from the top and the bottom to the system. The wavefield of a single source can be expressed by the sum of a power series as shown in Nakata et al. [28]. Expanding from this, the wavefield of two sources with the configuration in Figure 5 is a superposition of their individual wavefields. That is,

$$U(z, \omega) = \frac{\begin{array}{l} S_1(\omega)(e^{z(ik - \gamma|k|)} + R_2 e^{(2H-z)(ik-\gamma|k|)}) + \\ S_2(\omega)(e^{(H-z)(ik-\gamma|k|)} + R_1 e^{(H+z)(ik-\gamma|k|)}) \end{array}}{1 - R_1 R_2 e^{2H(ik-\gamma|k|)}} \quad , \qquad (3)$$

where $z$ is depth, $\omega$ is the angular frequency, $i$ = the imaginary number, $k$ is the wave number, $H$ is the length of the structure, and $\gamma$ is the attenuation factor where $\gamma = \frac{1}{2Q}$ [37], $Q$ is the quality factor, $S_1$ and $S_2$ denote the spectrum of the two sources terms, and $R_1$ and $R_2$ are the reflection coefficients of the top and bottom reflectors, respectively. In the nominator, $e^{z(ik-\gamma|k|)}$ and $R_2 e^{(2H-z)(ik-\gamma|k|)}$ are the direct wave and the first reflection for $S_1$, while $e^{(H-z)(ik-\gamma|k|)}$ and $R_1 e^{(H+z)(ik-\gamma|k|)}$ are those for $S_2$. Their amplitudes are scaled by the attenuation terms that involve $\gamma$. The $R_1 R_2 e^{2H(ik-\gamma|k|)}$ term in the denominator is the common ratio in the power series representing higher-order reverberations between two reflectors.

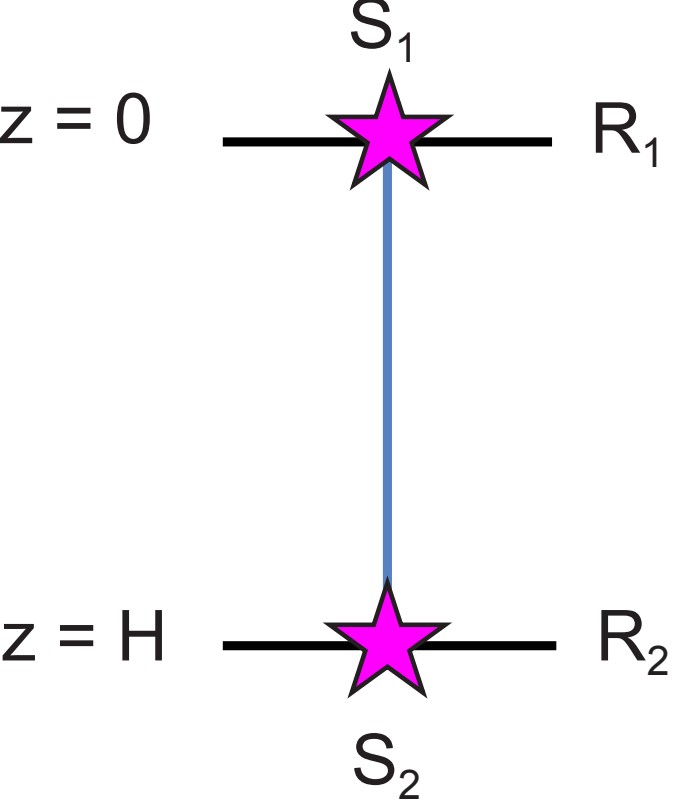

**Figure 5.** **Model 1**: A simple 1D string model used to simulate the deconvolved wavefields in Figure 4. The model has a string with a line of receivers on it (blue line) bounded by two reflectors at $z = 0$ ($R_1$) and $z = H$ ($R_2$). The two sources are located at $z = 0$ ($S_1$) and $z = H$ ($S_2$) (magenta stars).

We simulated the deconvolved wavefields using Equations (2) and (3) and compared them with the observed deconvolved wavefields (Figure 4a–f). After a series of parameter tests shown in Appendix A.1, we set all source terms to be mutually uncorrelated with their cross-correlation coefficient $cc = 0.01$. This choice was because correlated sources would generate simultaneous direct waves from the virtual source, as shown in Appendix A.1, which we do not observe in the wavefield. Other parameters used are $Q = 500$, $\omega/k = 4600$ m/s and $\varepsilon = 0.0001\%$, $R_1 = R_2 = 0.9$ for Figure 4a–d, and $R_1 = R_2 = 0.5$ for Figure 4e,f. These choices are based on the low attenuation across depth, apparent velocity of the signal, and high reflectivity at the boundaries in the observed data.

Figure 4 shows that we can reproduce the wavefields using Model 1. In Figure 4a–c, the same time window is examined using three different virtual sources (the red lines). The dominant waves exhibit symmetry between causal and acausal times (i.e., left and right to the blue lines), regardless of the virtual source. To achieve this symmetry regardless of the virtual source, the two sources in Model 1 need to have comparable amplitudes

(Appendix A.2). Figure 4d–f show that the model can also reproduce the three special cases observed in the data. In Figure 4d, the multiples are much weaker than in Figure 4a. We reproduced this case by letting the source $S_1$ at the depth of the virtual source be much weaker than $S_2$, the source at the opposite end of the interval. In Figure 4e,f, the dominant waves are asymmetric with only causal waves. We reproduced these cases by minimizing the amplitude of one of the sources and using the main source as the virtual source. Hence, we can explain these special cases with unequal amplitudes of $S_1$ and $S_2$. In Appendix A.2, we analyze the effect of varying relative source amplitudes.

Some observed deconvolved wavefields suggest a more complicated model (Figure 6). The observed wavefield in the top panel of Figure 6a shows a reflector at near 90–100 m. We reproduce this wavefield using Model 2 shown in Figure 6b. In Model 2, we add an additional source $S_{1a}$ co-located with $S_1$ at $z = 0$ (the dark blue star). This additional source generates waves that propagate between $z = 0$ and $z = H/2$ (the dark blue line). A reflector $R_3$ at $z = H/2$ acts as a lower boundary for this wave. The high RMS amplitude near 90–100 m in Figure 2d supports this model.

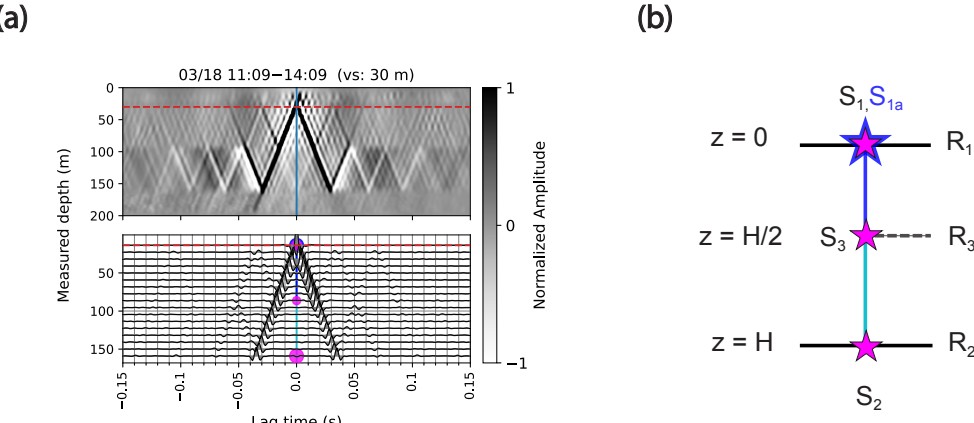

**(a)** **(b)**

**Figure 6.** A deconvolved wavefield (**a**) that can be explained by a more complicated model, (**b**) Model 2. In (**b**), we add two additional sources ($S_{1a}$ and $S_3$), and a middle reflector ($R_3$). $S_{1a}$ is at $z = 0$ (blue star) and propagates only between $z = 0$–$H/2$ (the dark blue line). $S_3$ is at $z = H/2$ and propagates between $z = 0$–$H$ (both the dark blue and the cyan lines). $R_3$ is at $z = H/2$ (dashed black line).

### 3.3. Time-Lapse Changes of Wave Velocities

In this section, we analyze the velocity evolution using the extracted wave. The deconvolved wavefields are calculated with 1 min time windows and 50% overlap, and are stacked over 1 h. We calculated deconvolved wavefields with the virtual source at 180 m and measured the arrival times by picking the peaks of the upgoing direct wave within 70–120 m. The signals are the most consistent over the eight days between this depth range. We calculate the velocities for a channel by dividing the measured travel length (between the source channel and the target channel) by the chosen arrival time. In Figure 7, we plot the estimated velocities against measured depth, observation time, temperature, and pressure. Each gray dot is a velocity measurement at a channel. In general, the velocity of this signal is at around 3600–5000 m/s. This velocity range is much higher than that of the local formation ($V_p$=1000–2500 m/s; Parker et al. [38], Parker et al. [38]). The velocities are closer to the compressional velocity of steel (5000–5250 m/s; Haynes [39]). The waves likely propagate in the stainless steel DAS cable jacket or the steel well-casing, as Miller et al. [14] suggested.

Figure 7a shows the velocities across the measured depth of 70–120 m. The velocities show a slight decreasing trend of −6.6 m/s per meter, which reflects the negative temperature-velocity dependency in Figure 7c, since the temperature increases with depth at this depth range (Figure 2b). The velocity variations (the width of the blue shade) are

larger near 72 m and 100 m. This larger variation potentially indicates poor coupling of the DAS cable or it may be related to the complicated structure and additional source observed in Figure 6.

Figure 7b shows the velocity evolution over the eight days. Early on 18 March, the mean velocity suddenly rose from 4100 to 4700 m/s. The velocity fell back to 4100 m/s before late 19 March. It fluctuated between 4100–4300 m/s for the remaining of the time. The rise of velocity during 18–19 March is likely associated with disturbances in the borehole. This disturbance was caused by depressurization boiling due to the initial pressure drop related to increasing operation intensity [24] (Figure 2a). During this time, the DAS data also had a high DC level (Figure 2c).

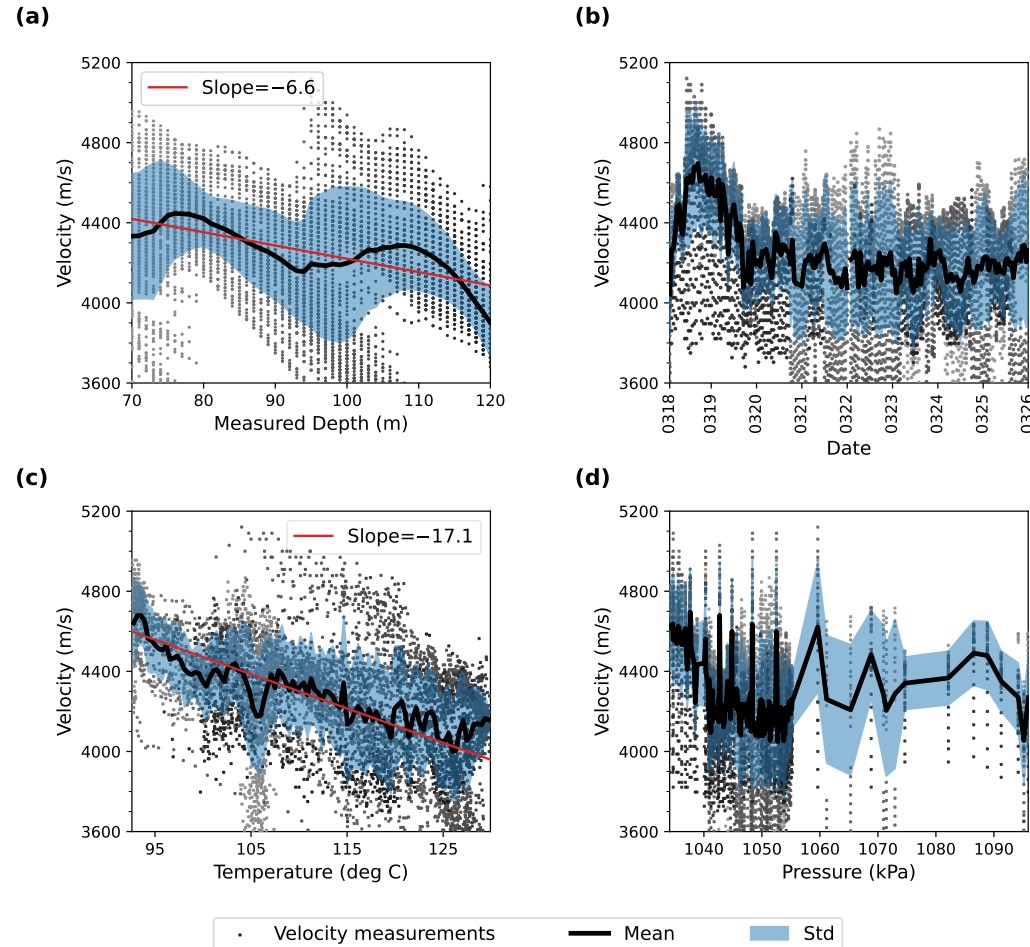

**Figure 7.** The velocity measurements of the extracted wave versus (**a**) measured depth, (**b**) observation time, (**c**) temperature, and (**d**) pressure. Each gray dot is a velocity measurement at one channel in the 1-hour stacked deconvolved wavefields. Deeper channels are distinguished by gradually darker colors. The black curves are the mean values. The blue shades mark one standard deviation above and below the mean values. The red lines in (**a**,**c**) show the trend of the linear fit estimated from the gray dots.

Figure 7c shows that the velocity decreases with increasing temperature with a slope of −17.1 m/s/°C. This temperature sensitivity is much higher than that measured in the lab for pure steel material (−0.5 m/s/°C; Mott [40]; Droney et al. [41]). We have two possible explanations for this. If the waves propagate in the DAS cable jacket, then this higher sensitivity might suggest the cable or the fiber inside being subjected to the high temperature. We note that the DAS fiber is rated to 150 °C, while the highest temperature in the borehole is beyond 160 °C (Figure 2b). On the other hand, if the waves propagate in the well-casing, it suggests that the casing might have higher sensitivity to temperature. In

Figure 7d, we do not observe an obvious relationship between velocity and pressure due to the lack of samples at higher pressure.

### 3.4. Normal-Mode Analysis

The deconvolved wavefield of a vibrating 1D structure can be written as the summation of normal modes [28,29]. This is observed in our results. Figure 8 shows the amplitude spectrum of one of the deconvolved wavefields we used for time-lapse analysis. The normal modes of the signal are clearly decomposed from 10 Hz to over 200 Hz. The frequency interval between the different modes is about 18 Hz and consistent over all modes, as expected.

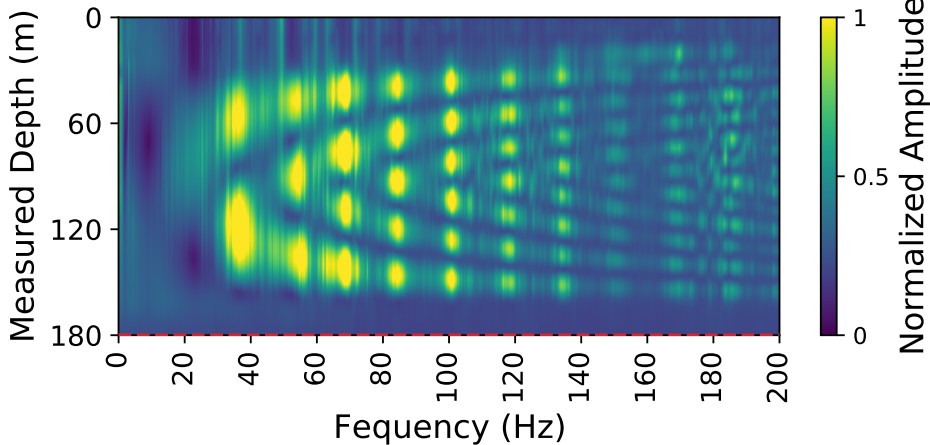

**Figure 8.** The deconvolved amplitude spectrum of DASV. It is calculated using 1 min time window and 50% overlap, and stacked over 1 h. The reverberating waves are clearly decomposed into normal modes with ∼18 Hz frequency interval. The red dashed line marks the virtual source at 180 m.

The system has closed boundaries on both ends. The top boundary is due to the free surface that behaves as closed boundary for P-wave multiples. The bottom boundary is because of the deconvolution modifying the boundary condition to a clamped boundary (a delta function) at the virtual source [26]. For this system, the wavelength of mode *m* is [42]

$$\lambda_m = \frac{2H}{m} \quad , \tag{4}$$

where *H* is the length of the system. Hence, the phase velocity for mode *m* is

$$c_m = \lambda_m f_m = \frac{2H f_m}{m} \quad , \tag{5}$$

where $f_m$ is the mode frequency. We estimated $f_m$ and $H$ in the hourly stacked amplitude spectrum at 6 or 7 am on each day. This is a time with relatively high signal-to-noise ratio. We focused on the second (∼38 Hz), the third (∼55 Hz), and the fourth (∼71 Hz) modes, since these three modes are the most significant. We picked $f_m$ at the peak amplitude of each mode. We estimated $H$ by picking the starting and ending depths of the mode. Then, we calculated the phase velocity using Equation (5).

The temporal variations of mode frequency, system length, and phase velocity are shown in Figure 9. The mode frequencies increase slightly with time, while the system length decreases with time (Figure 9a). The velocities estimated using higher modes are lower, suggesting a negative frequency–velocity dependency. Hence, the velocities are dispersive [43]. In general, the three modes show similar trends: the velocities increased for the first few days before 20 March, and then they continuously decreased until the end of the analysis period.

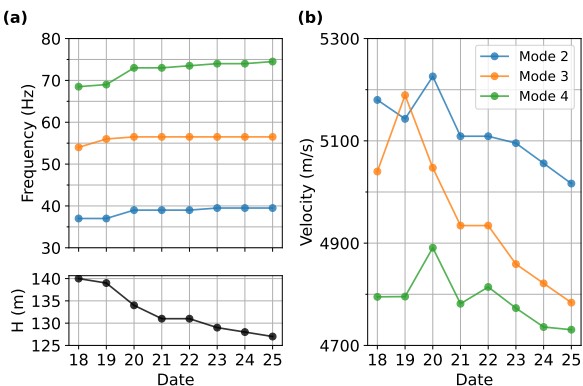

**Figure 9.** Parameter variations of the amplitude spectra. (**a**) The peak frequencies ($f_m$) for the second, third, and fourth modes increase over time. The system length ($H$) decreases over time. (**b**) The phase velocities of the wave calculated using Equation (5).

## 4. Discussions

We extract coherent waves in the borehole DASV data using deconvolution seismic interferometry (Figure 4). The extracted waves are the ringing of the DAS cable and the well casing based on the velocity (Figures 7 and 9b). These are caused by poor coupling between the cable to the well or between the well and the formation. By using different virtual sources, we examine the wavefields that satisfy different boundary conditions. A simple model with two sources and two reflectors (Model 1) can explain the deconvolved wavefields (Figure 4). Some wavefields exhibit more complexity and suggest a more sophisticated model (Model 2; Figure 6). We use numerical simulations to qualitatively reproduce the direct waves and the multiples in the deconvolved wavefields. In Model 1, the reflectors are associated with free surface and potential casing defects [14,24]. In Model 2, the added sources and the wave trapped in the upper half of the system (the dark blue line in Figure 6b) potentially suggest a separate vibration of the outer casing on the first 87 m of the well [14]. In fact, the actual conditions might be even more complicated and we note that the solutions are not unique. Therefore, this model may not be appropriate if one wants to analyze the inverted full wavefield. However, this model may be the simplest that can well explain our extracted waves.

One important feature of the wavefields in Figure 4a–c is the symmetry. According to Nakata and Snieder [30], to have symmetry between the causal and acausal times for all virtual sources in this model there must be more than one source. In Appendices A.2 and A.3, we find that asymmetry is produced by uneven amplitudes of the sources or uneven reflectivities of the reflectors. The effect of the former on asymmetry is more dominant than the latter. We reproduce the symmetry in simulations by having two sources with comparable amplitudes and reflectors with identical reflectivities.

The main energy sources in this system are borehole processes, surface operations, and traffic noises. The relative amplitudes of these sources change over time, resulting in different observed cases of deconvolved wavefields in Figure 4. This noise variation can generate the asymmetry discussed above. The borehole processes include depressurization boiling and fluid exchange activities at potential casing defects [14,24]. These processes are the most intense during the initial pressure drop. Hence, the deconvolved wavefield shows strong upgoing waves during this time (Figure 4e). The surface operations include site activities and vibroseismic experiments that were conducted 10 h a day. During these vibroseis experiments, we observed strong downgoing waves (Figure 4f). In addition, the interstate highway on the north-western side of the survey region provided traffic noises as a general energy source of the extracted wave [44].

We separated the direct waves and the multiples of the ringing signals by using the base of this system as the virtual source. We tracked the velocity variations of the direct waves. The dense spatial sampling of DAS allowed us to observe the trend of velocity variations along depth, time, temperature, and pressure within a 50 m interval (Figure 7).

The velocity variations provide potential information for coupling conditions along the cable and the parameters to which these ringing signals are sensitive. In Figure 7a, the depth with large velocity variations might suggest poor coupling or presence of energy sources or complex structures. The latter is also supported by Model 2, which explains occasional variation of the wavefields (Figure 6). As discussed in Section 3.3, the temporal correlation between the high velocity and the time of large borehole disturbances suggests that the ringings are sensitive to these disturbances (Figure 7b). Finally, in Figure 7c, the decreasing velocity with temperature indicates that the DAS cable or the casing is potentially sensitive to high temperature. Hence, by monitoring the extracted waves, we can gain information on the medium in which the waves propagate.

The velocities estimated by choosing arrival times of the propagating wave (Figures 7) are slightly slower than the normal-mode method (Figure 9). This is because the normal-mode analysis is done in the lower frequency modes which have higher velocities (Figure 9b), whereas the propagating waves contain all frequencies. The frequency-dependent velocities from the normal-mode analysis are potentially useful to obtain attenuation and structures at different distances from the well. However, in this case, since the coupling (either between the DAS cable to the casing or between the casing to the formation) was poor, the dispersion relation is less sensitive to the structure. Instead, the negative frequency–velocity relation might be caused by the casing and fluid in the borehole, but we require a further experiment to understand the dispersion of the waves.

We note that this deconvolution method can be useful for monitoring changes in the reservoir. In Appendix B, we show the signals we extracted on the lower portion of the DASV cable (below 200 m). We were able to obtain signals during some of the vibroseis experiments. However, the poor signal-to-noise ratio prevented us from analyzing the time-lapse changes with good precision. If the coupling were better, the signal-to-noise ratio would be improved and we could have signals outside of the vibroseismic experiment times. Then, we could apply the similar time-lapse velocity analysis on the obtained signals and infer for reservoir property changes.

## 5. Conclusions

We used deconvolution seismic interferometry to analyze the reverberations in distributed acoustic sensor (DAS) in the borehole. This method is useful for understanding complicated wavefields. The waves we extracted on the shallower part of the borehole are the cable and casing ringing. By examining the wavefield at different boundary conditions, we can qualitatively interpret the system using a simple 1D string model. An important observation is the symmetry of the wavefield. The keys to explain our observations are the source correlations, relative source amplitudes, and reflection coefficients in this system. The deconvolution method also allows us to separate the direct waves and the multiples, and track the velocity changes of the direct waves over time. The velocity experienced a rise during the initial pressure drop that was associated with increasing operation intensity. The velocity decreased with increasing temperature and depth. The velocity sensitivity to temperature is higher in our results than that for pure steel measured in the lab. This suggests that the DAS cable or the well-casing is potentially affected by the high temperature. The technique proposed here can be applied to many different borehole DAS applications. These applications include diagnosis of the condition of the casing structure and monitoring changes in reservoir properties. For the latter, we need better coupling than simple friction in the vertical borehole in order to obtain more energy from the formation.

**Author Contributions:** Conceptualization, H.C. and N.N.; methodology, H.C. and N.N.; software, H.C.; validation, H.C.; formal analysis, H.C.; investigation, H.C.; resources, N.N.; data curation, H.C.; writing—original draft preparation, H.C.; writing—review and editing, N.N.; visualization, H.C.; supervision, N.N.; project administration, N.N.; funding acquisition, N.N. All authors have read and agreed to the published version of the manuscript.

**Funding:** This study was supported by the Japan Oil, Gas, and Metals National Corporation and the MIT Indonesia Seed Fund.

**Institutional Review Board Statement:** Not applicable.

**Informed Consent Statement:** Not applicable.

**Data Availability Statement:** Publicly available data sets were analyzed in this study. This data can be found at the DOE Geothermal Data Registry (GDR https://gdr.openei.org/; last accessed on 4 November 2021).

**Acknowledgments:** We appreciate the advice from Douglas Miller. We thank the useful feedback from anonymous reviewers, Michal Chamarczuk, and Deyan Draganov. They helped to improve this manuscript significantly. The computing for this project was partially performed at the OU Supercomputing Center for Education and Research (OSCER) at the University of Oklahoma (OU). This study was supported by the Japan Oil, Gas and Metals National Corporation and the MIT Indonesia seed funds.

**Conflicts of Interest:** The authors declare no conflict of interest.

## Abbreviations

The following abbreviations are used in this manuscript:

| DAS | Distributed acoustic sensing |
| DTS | Distributed temperature sensing |
| RMS | Root-mean-square |

## Appendix A. Varying Model Parameters

In this section, we simulate the deconvolved wavefields with varying parameters in Model 1. The parameters in Section 3.2 are based on the analysis here.

### *Appendix A.1. The Effect of Source Correlation*

We varied the correlation coefficient between $S_1$ and $S_2$ from 0.01 (uncorrelated), 0.5 (partially correlated), to 0.99 (highly correlated). To generate synthetic data with a certain degree of correlation, we first generated random, normalized data time-series and put them in rows to form matrix $A$. We build a covariance matrix $R$ with the desired correlation coefficient ($cc$) on the non-diagonals and 1 on the diagonals. Then, we use the Cholesky decomposition to calculate matrix $C$ such that $CC^T = R$. Multiplying $A$ with $C$ gives a new matrix where the $cc$ between each row is as desired.

Figure A1 shows the simulation results when the $cc$ equals 0.01, 0.5, and 0.99. When the two sources are not correlated (Figure A1a), only the virtual source emits waves from time zero. When the two sources are correlated (Figure A1b,c), the correlated source emits another set of waves in addition to that from the virtual source. The higher the correlation, the larger the amplitudes of those simultaneous direct waves. Since we do not observe these simultaneous direct waves in the data, we set $cc = 0.01$ in the simulations in Figure 4.

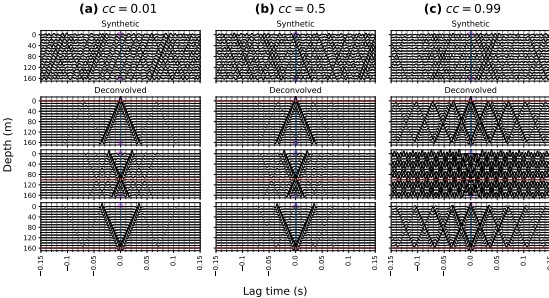

**Figure A1.** Simulation of the deconvolved wavefield with varying source correlation: (**a**) not correlated ($cc = 0.01$), (**b**) partially correlated ($cc = 0.5$), and (**c**) highly correlated ($cc = 0.99$). The magenta balls are the actual sources. The virtual sources are marked by the red dashed lines. Other parameters are default values: $|S_1|/|S_2| = 1$, $R_1 = R_2 = 0.5$, $Q = 500$, $\omega/k = 4600$ m/s, and $\varepsilon = 0.0001\%$.

*Appendix A.2. The Effect of Relative Source Amplitude*

We vary the relative amplitude of the two sources ($|S_1|/|S_2|$) from 0.1, 1, to 10. When $|S_1|/|S_2| = 1$ (Figure A2b), the relative amplitudes on the causal and acausal axes are well matched, regardless of the depth of the virtual source. The wavefields are symmetric. When $|S_1|/|S_2| = 0.1$ (Figure A2a), the waves at causal times have larger amplitude for channels above the virtual source (the red dashed lines), whereas for channels below the virtual source, the waves at acausal times have larger amplitudes. When $|S_1|/|S_2| = 10$ (Figure A2c), the patterns reverse.

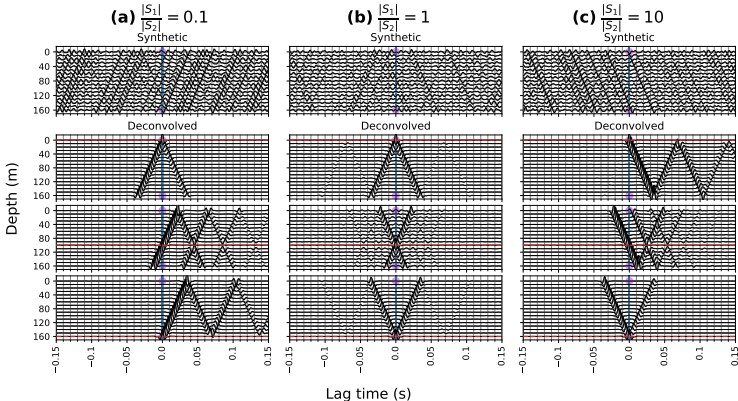

**Figure A2.** Simulation of the deconvolved wavefield with varying relative source amplitude $|S_1|/|S_2|$: (**a**) strong source from below ($|S_1|/|S_2| = 0.1$), (**b**) sources with equal amplitudes ($|S_1|/|S_2| = 1$), and (**c**) strong source from above ($|S_1|/|S_2| = 10$). Other parameters are default values: $R_1 = R_2 = 0.5$, $cc = 0.01$, $Q = 500$, $\omega/k = 4600$ m/s, and $\varepsilon = 0.0001\%$.

When one of the sources is dominant (Figure A2a,c), the deconvolved wavefields approach the one-source cases. This is predicted by the equations. Based on Equations (1) and (3), the deconvolved wavefield using a virtual source at $z_a$ ($0 \leq z_a \leq H$) can be written as

$$D(z, z_a, \omega) = \frac{\begin{aligned}&\tfrac{S_1}{S_2}\left(e^{(z+z_a-H)(ik-\gamma|k|)} + R_2 e^{(H-z+z_a)(ik-\gamma|k|)}\right)+\\ &\left(e^{(z_a-z)(ik-\gamma|k|)} + R_1 e^{(z_a+z)(ik-\gamma|k|)}\right)\end{aligned}}{\begin{aligned}&\tfrac{S_1}{S_2}\left(e^{(H+2z_a)(ik-\gamma|k|)} + R_2 e^{H(ik-\gamma|k|)}\right)+\\ &\left(1 + R_1 e^{2z_a(ik-\gamma|k|)}\right)\end{aligned}} \tag{A1}$$

$$= \frac{\begin{aligned}&\left(e^{(z-z_a)(ik-\gamma|k|)} + R_2 e^{(2H-z-z_a)(ik-\gamma|k|)}\right)+\\ &\tfrac{S_2}{S_1}\left(e^{(H-z-z_a)(ik-\gamma|k|)} + R_1 e^{(H+z-z_a)(ik-\gamma|k|)}\right)\end{aligned}}{\begin{aligned}&\left(1 + R_2 e^{2(H-z_a)(ik-\gamma|k|)}\right)+\\ &\tfrac{S_2}{S_1}\left(e^{(H-2z_a)(ik-\gamma|k|)} + R_1 e^{H(ik-\gamma|k|)}\right)\end{aligned}}. \tag{A2}$$

In the case when $|S_1|/|S_2| \approx 0$ and $z_a = H$, Equation (A1) becomes

$$D(z, H, \omega) = \sum_{n=0}^{\infty}(-1)^n \times$$
$$\left(R_1^n e^{((2n+1)H-z)(ik-\gamma|k|)} + R_1^{n+1} e^{((2n+1)H+z)(ik-\gamma|k|)}\right). \tag{A3}$$

Similarly, in the case when $|S_2|/|S_1| \approx 0$ and $z_a = 0$, Equation (A2) becomes the infinite series of Equation 9 in Nakata et al. [28], which has similar form. The *ik* terms in the exponents in Equation (A3) and Equation 9 in Nakata et al. [28] are all positive. Hence, the wavefield is asymmetric.

*Appendix A.3. The Effect of Reflection Coefficients*

We varied the reflection coefficient on the top boundary ($R_1$) from 0.01, 0.5, to 0.99. The effect is not obvious in the mathematical notation, but is observable in the simulated wavefields (Figure A3). When $R_1$ becomes larger, for channels above the virtual source, the acausal waves are enhanced, whereas for channels below the virtual source, the causal waves are enhanced.

This phenomenon of a larger $R_1$ is the opposite of the effect of a larger $S_1$. That is, based on Appendix A.2, if $|S_1|$ increases relative to $|S_2|$, we expect the causal waves to be enhanced for the channel above the virtual source, while the acausal waves should be enhanced for the channel below the virtual source. Hence, the relative amplitudes between the causal and acausal axes can be affected by both the relative source amplitude and the reflection coefficients. However, the influence of the reflection coefficient on the symmetry is subtle. We set $R_1 = R_2 = 0.9$ in the simulations in Figure 4.

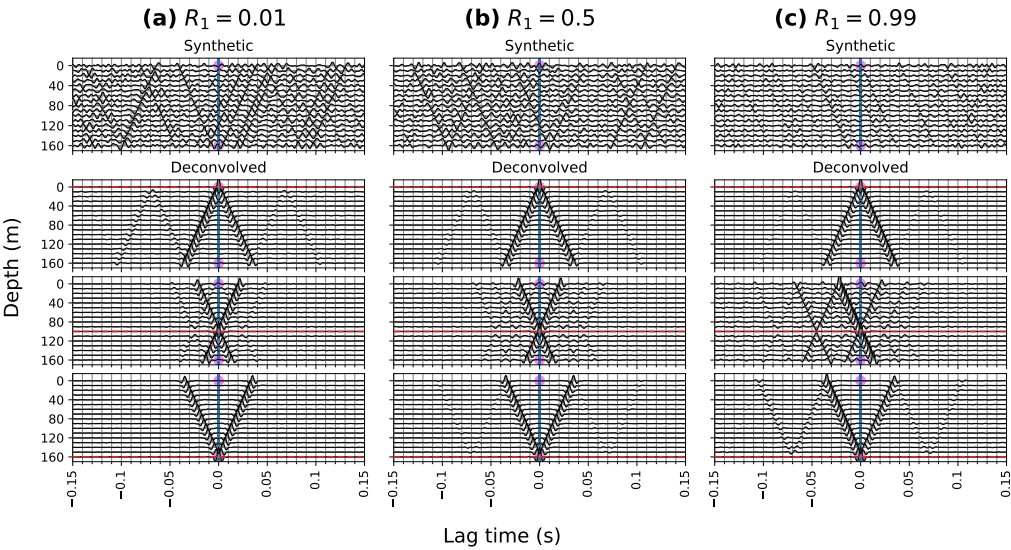

**Figure A3.** Simulation of the deconvolved wavefield with varying reflection coefficients of the top reflector ($R_1$): (**a**) low reflectivity ($R_1 = 0.01$), (**b**) intermediate reflectivity ($R_1 = 0.5$), and (**c**) high reflectivity ($R_1 = 0.99$). Other parameters are default values: $|S_2|/|S_1| = 1$, $R_2 = 0.5$, $cc = 0.01$, $Q = 500$, $\omega/k = 4600$ m/s, and $\varepsilon = 0.0001\%$.

**Appendix B. Deconvolved Wavefields at the Lower Part below 200 M**

We extracted coherent waves from the formations. However, these waves were only present during the times when the vibroseis shots were close to the DASV well. Figure A4 shows three sets of waves we extracted in the depth range 165–300 m. This deconvolved wavefield is calculated using 30 min time windows and 50% overlap, and is stacked over 3 h.

The first two signals (Figure A4a,b) traveled downward with the apparent velocities of 2100 m/s (green dashed lines) and 1100 m/s (pink dashed line). The $V_p/V_s$ ratio is 1.91. This is consistent for shallow formations in Brady, which consists of volcanic sediments, limestone, lacustrine sediments, and geothermal features such as carbonate tufa [45]. The measured velocities are close to previously estimated local velocities ($V_p = 2300$ m/s and $V_s = 1200$ m/s; Parker et al. [38]; Matzel et al. [44]). The slower apparent velocities might be due to incident angles. Potentially, we could estimate the time-lapse changes by measuring the relative velocity changes of these waves. However, the poor coupling condition prevented us from obtaining more scattering energy.

The third signal (Figure A4c) had an apparent velocity of 1400 m/s (the yellow dash line) and propagated upward. It was weaker than the first two signals. The source of this signal could be a reflection from nearby faults or bedding planes [25,45]. The most likely

case is that there was an upgoing wave. However, we cannot identify the reflection point due to the limited amount of good-quality data.

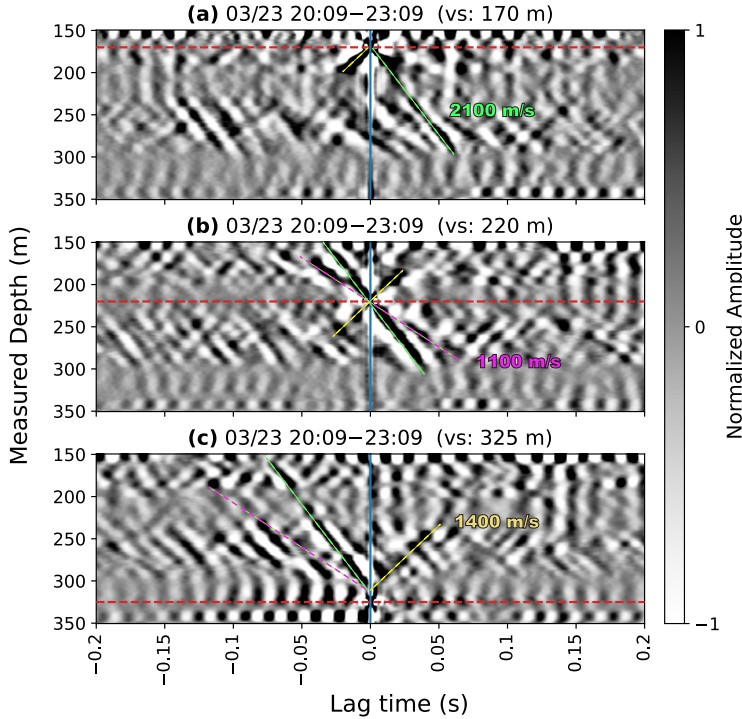

**Figure A4.** Three sets of waves are extracted between 150–350 m. (**a**–**c**) show these waves in the deconvolved wavefields using different virtual sources (red dashed lines) at the same time. The green and magenta dashed lines are two downgoing waves. The yellow dashed lines are the upgoing wave. These signals were only present during times where a vibroseis truck was operating at nearby sites (100–600 m away). They are calculated using 30 min time windows and 50% overlap, and are stacked over 3 h.

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
