# Peer review of "Investigation of Time-Lapse Changes with DAS Borehole Data at the Brady Geothermal Field Using Deconvolution Interferometry"

_remotesensing, doi:10.3390/rs14010185_

Round 1
Reviewer 1 Report
After modification by the authors, this paper can fully display the research results of the authors.
The authors use deconvolution method to examine the wavefield. It allows us to interpret the wavefield with a simple model. Furthermore, it separates the direct waves and the multiples and simplifies the wavefields. This makes time-lapse monitoring easier to implement. At last but note least, we also have some advice to authors. Firstly, it is best to add a flowchart to the methods section. Secondly, the conclusion in Abstract section need to be supported by specific experimental data. Finally, the discussion section needs to be augmented by some more specific experiments.
Reviewer 2 Report
The paper is interesting and describes a serious problem faced by anyone who would like to use DAS data where the coupling between the DAS cable and the borehole casing is of varying quality. However, there are some points that should be addressed.
I would suggest that the phenomenon the authors are describing is “cable ringing” and not casing ringing. “Casing ringing” is caused by energy traveling in the casing where the casing is free to move independently because of insufficient cement bonding to the formation. Similarly, insufficient coupling to the casing of the cable with the embedded fiber, allows the cable to move independent of the casing.
Like the authors, I believe that Miller et al. describe this “cable ringing” as a “guitar string” effect - imagining energizing a string that is free to move between two fixed points. The authors describe the “cable ringing” mathematically in terms of a vibrating string of finite length. This string has a source of energy at either end, and either end of the string also generates reflections. With such a model, one may have expected the authors to attempt to estimate the limited number of parameters directly from the data using a sort of least-squares estimation process. One may speculate that a good description of the “cable ringing” noise may allow this to be estimated and removed to reveal data that are more interesting to the owner of the well. This might have allowed the authors to do an analysis along the shallower part of the well, like what they do in Appendix B for the deeper part of the well. However, as it now stands, I do not consider the material in Appendix B to be essential to the main topic of this paper, which appears to be describing the “cable ringing” phenomenon.
What is the standard deviation on the slope of -17.1 m/s/â—¦C (in lines 234-235)? Was this slope estimated from the raw (temperature, velocity) data tuples, or from the average velocity for each temperature?
In Figure 8, the velocities of the dominant modes vary between about 4730 m/s and 5220 m/s. Is this consistent with the range of velocities shown in Figures 6?
Specific comments, mostly on formulation and language:
Line 7: Replace “tirelessly” with (e.g.) “frequently”.
Line 16: Replace “the the” by “the”.
Line 25: Replace “stream” by “string” or “array”.
Line 41: The terms “artificial stripes” is not very descriptive. This could be replaced by “noise that occurs simultaneously and same amplitude on all sensor channels”.
Lines 41-42: Replace “casing ringing” with “cable ringing”.
Line 69: Replace “took this advantage” by “used this”.
Line 70: Replace “the building” with “a building”.
Line 74: Replace “with a simple model” with “using a simple model”.
Line 77: Replace “In below” by “In the following”.
Line 80: Replace “this signal” by “the recorded signal”.
Line 87: Replace “that spans about 380 m” with “that is about 380 m deep”(?)
Line 97: Replace “wellhead” with “wellheads”.
Line 125: Replace “retrieved” by “retrieve”.
Line 126-127: Replace “monthly and annually shear wave velocity changes between the” with “monthly and annual changes in shear-wave velocity using”.
Line 167: Replace “i is the imaginary number” with “i = ”.
Line 169: Define “Q”.
Line 180: Replace “source” with “sources” (two occurrences).
Line 186: Do you mean to say “with two sources on both boundaries”, or do tou rather mean “with one source on either boundary” (which would be consistent with Figure 4, and equation 3?
Line 191: Replace “Figure 3d-3f show the model” with “Figure 3d-3f show that the model”.
Line 193-194: I would suggest replacing “We reproduce it by putting the dominant source on the other side of the system to the source” with “We reproduce it by letting the source S1 at the depth of the virtual source, be much weaker than S2, the source at the opposite end of the interval”.
Line 195: Replace “We reproduce it…” with “We reproduce this…”.
Line 211: Replace “between” with “in” or “within”.
Line 215: Descriptive inconsistency between text and figure caption. Referring to Figure 6, the text says “grey dot”, whereas the caption of Figure 6 says ”black dot”.
Line 376: Replace “1s” with “1” (??).
Line 417: Replace “these waves only present” with “these waves are only present”.
Line 424: Replace “is consisted of” with “consists of”.
Lines 427-428. The statement “The slower apparent velocities might be due to incident angles” is wrong. The apparent velocities are always larger than the formation velocities.
Lines 429-430: Take out “if we could extract them more often in time”.
Page 18: In the author list of references 2, 3, 4, 6, 39, 46, the term “other” appear, I assume by error?

Author Response
Please see the attachment.

This manuscript is a resubmission of an earlier submission. The following is a list of the peer review reports and author responses from that submission.
Round 1
Reviewer 1 Report
The paper presents a deconvolution method on the DASV data from 2 of the 4 week PoroTomo survey. Overall, it is difficult to decipher the contribution of the work, and clearly see the connection between what you learned from the synthetic modeling and how it relates the the real data results.
- I believe that your synthetic modeling should be better integrated into the main paper. Can you identify and put the synthetic cases that enlighten or "match" the real data deconvolution results into the main body? It is the job of the author to make the connections and your interpretations obvious to the reader. Right now I'm not clear which scenarios were observed more and what the take away's are relating to the sources.
- Also, be clear about your contributions in this work. Is it simply an application of previously defined methodology to new data? That's fine, but then emphasize other contributions.
- How do your findings differ from the Miller study. I walk away from reading the work without clearly understanding this.
- One angle could be how this could be a cheap well integrity diagnostic technique (very important for reducing greenhouse gas leaks, etc).
- Include this in introduction.
- The organization of some paragraphs are muddled. Make sure you understand the "thesis" of each paragraph and make sure the rest of the paragraph is succinctly and clearly within that thesis.
Marked-up pdf attached.

Author Response
Dear reviewer:
Thank you very much for your valuable review comments. We have revised the manuscript based on your suggestions. Please see the attachment for the point-by-point responses to your comments and the revised manuscript that marks all the revisions (Note: All the line numbers refer to those in the clean version of the pdf, not the pdf that marks the difference).
Below is the summary of the revisions based on comments from multiple reviewers:
Here are two major structural revisions:
- To improve the integration between Appendix B and the main text, we
- Moved the main results of the analytical simulation (originally in Appendix B) to the main text (Section 3.1.3).
- Replaced Figure 3 with 1-to-1 comparisons between the observed and simulated wavefield.
- For interpretation of deconvolved wavefields, we
- Removed the description of the wavefield in discussion (originally in line 322-348), Appendix B.5, and Figure B.5.
- Used a second model (added Figure 5) to support our claims in Section 3.1.3 instead.
Here are some major content revisions of some sections (also commented in the pdf file):
- Abstract: Rewrote the abstract to clearly emphasize our novelty.
- Section 1 (Introduction): Made a better connection between previous studies and this study.
- Section 2 (Data): Made a clearer list that explains features in Figure 2 that relate to later interpretations.
- Section 3 (Methods and analysis results):
- 3.1.1 Added reasons for the choice of method. Simplified the description of the time windows.
- 3.1.3 Added 1-to-1 comparisons between the observed and simulated wavefield and made better connections between analytical and numerical simulation results using the replaced Figure 3.
- 3.2 Interpreted Figure 6 here and discuss them later in section 4.
- The entire Section 4 (Discussion):
- Reorganized the paragraphs and emphasized discussing the main results of this study.
- The entire Section 5 (Conclusion): Rewrote the paragraphs based on the revised version of Section 4.
- Appendix A (Deconvolved wavefields at the lower part): Added explanations of our interpretation.
- Appendix B (Varying modeling parameters): Integrated into the main text by
- Moved the original B1 to 3.1.3.
- Removed the original B5 and explained it using model 2 in 3.1.3.
Please refer to the highlighted pdf for details of these revisions. Thank you very much.

Reviewer 2 Report
The paper uses deconvolution interferometry to extract useful information from DAS Borehole data, and monitor time-lapse changes of the propagation media and external sources, such as the integrity of the well casing. Using this method, people can understand the situation under drilling better. The research approach and results are sound, and I have only minor comments.
1-Page 2, Line 86. “green cross in Figure 1…”, ‘cross’ should be ‘dot’?
2-Figure 1. The blue triangles are too small so that it seems like black triangles.
3-Page 3, Line 111. “The maximum temperature is around 160â—¦C.” But in line 91 of page 3, “…which is tested to be resilient up to 150â—¦C.” Will high temperature affect the result?
4-Page 5, Line 156. “These longer time intervals are required for enhancing the SNR in the lower part.” And in line 149, it is said “A shorter tstep or longer tspan increases stack number and improve SNR.” What is the role of tstep, will the longer tstep reduce the SNR? (tstep = 0.5min → tstep = 15min)
5-Figure 3. “vs 10, vs 60, vs 120, vs 165”. Does the number behind ‘vs’ represent the depth of ‘vs’? It can be written more clearly.
6-Page 11, Line 340. “The top reflector seems to move from 10 to 35m at this stage (Figure 6a).” It is hard to understand and I can’t match “10 to 35m” with the figure.
Author Response

(The authors gave the same response as above.)

Reviewer 3 Report
This paper proposed a method to study the time-lapse changes with the DAS borehole data using deconvolution interferometry. Overall, this is not a well-organized paper. Here are my detailed comments:
[1] The Abstract section does not provide a complete and clear description of the content and effect of the research in this paper, so we suggest that the Abstract section be rewritten.
(2) The Introduction section does not provide a good overview of related studies, so it is difficult for readers to judge the innovativeness of this paper. The authors should add more relevant research status.
(3) The section of Method is poorly organized. The authors only introduced deconvolution interferometry without a detailed and specific introduction of how to combine it with the research in this paper.
(4) The conclusions based on the analysis shown in Figure 4 are hard to convince, and the relationship between the data and the pattern is not as straightforward as the author describes.
(5) The description of the experimental results also is subjective, and it is difficult to see the relevant data support.
(6) What is the relationship between the Appendix and this paper? Can they be combined?
(7) To sum up, I think the readability and innovation of the paper are not obvious, and the authors should show them more.
Author Response

(The authors gave the same response as above.)

Reviewer 4 Report
Dear authors, I found your manuscript very interesting, I especially liked modelling results your used to characterize the results retrieve from seismic interferometry. I think that technical part is correct, the manuscript is understandable, but could be improved by a proof reading. In an attached pdf I am giving detailed textual comments. Below, I am bringing up several points, that I think need clarification: 1. Page 5, line 144, "...for one correlogram...": you have not introduced the readers to the term "correlogram". Please define it before you use it, especially because you are now using deconvolution instead of correlation. You can mention that the numerator in equation 2 actually represents correlation. However, note that other authors (e.g., Schuster) have introduced and use "correlogram" to represent a correlation panel containing multiple correlated traces to be stacked over the available (noise or active) sources. You do not really mean that, so I would advise that you use another term here. Possibly deconvolution result? And then, after the summation over time panels, “final retrieved result”? 2. Figure 3: in the caption, explain the letter labels, i.e., (a) to (l), of the separate pictures. 3. Page 6, line 170, "...move inward and outward simultaneously...": what do you mean here by this? Up and down the borehole? Please, explain to the readers. 4. Page 6, "At 165 m, the wavefields have an abrupt change, below which we do not see the reverberations": I think you have to reformulate your statement here. To me it appears that there are reverberations in the right column in the lowest two rows, but they are of a lower-frequency and lower-amplitude character, which also appear to propagate with a different velocity. Or do I see something different? 5. Is there a connection between the gauge of 10 m and the top level at which you observe the reflection of the direct waves? Is it possible that the reflection is actually at the earth's surface, but that due to the gauge it appears at 10 m depth? 6. Page 7, line 217, "The system has closed boundary on both sides...": Please, explain this further. Why this is so? At the top there is the free surface, so that is ok. What is found at the bottom level? Is there a layer (or a structure) with a higher or lower velocity than the layer (or a structure) just above it? 7. Page 7, line 222, "We observe that the mode frequency and the system length (H) are changing during the analysis period": How do you observe that? Please, explain. 8. Conclusions: introduce the acronyms before using them. 9. Appendix A: You analyze three retrieved direct waves with apparent velocities of 2100 m/s (green line), 1100 m/s (pink), and 1400 m/s (yellow). If the vibroseis truck was operating at 100-600 m distance from the DAS borehole, then are these events not artefacts? Especially taking into account the shallow depths you are looking at? To retrieve correct arrivals, the source should be nearly vertically above the receivers, otherwise the retrieved would have indeed apparent velocities, but how could one interpret them with certainty to say that something is a P-wave, the second an S-wave, and the third a reflection? Or do you assume that the multiple scattering improves the situation? Please, elaborate on this. 10. Here and there you write "traveling waves". Please, use everywhere "propagating waves". 11. And the following is my personal preference. The method of retrieval of the Green’s function is accepted to be called seismic interferometry. Because you have submitted your manuscript to Remote Sensing, where the term interferometry is used more often in other context, I advise that you use "deconvolution seismic interferometry" or "seismic interferometry by deconvolution". In this way it is differentiated from other methods using interferometry. Or you can state in the beginning that you apply “seismic interferometry by deconvolution, which from now on is written as deconvolution interferometry”. With kind regards, Deyan Draganov

Author Response

(The authors gave the same response as above.)
